# Status and Epidemiology of Maize Lethal Necrotic Disease in Northern Tanzania

**DOI:** 10.3390/pathogens9010004

**Published:** 2019-12-18

**Authors:** Fatma Hussein Kiruwa, Samuel Mutiga, Joyce Njuguna, Eunice Machuka, Senait Senay, Tileye Feyissa, Patrick Alois Ndakidemi, Francesca Stomeo

**Affiliations:** 1Tanzania Agricultural Research Institute (TARI) Tengeru Sub Center, P.O. Box 1253, Arusha, Tanzania; 2Biosciences Eastern and Central Africa-International Livestock Research Institute (BecA-ILRI) Hub, P.O. Box 30709-00100, Nairobi, Kenya; S.Mutiga@cgiar.org (S.M.); joyceleyn@gmail.com (J.N.); E.Machuka@cgiar.org (E.M.); stomeofra@gmail.com (F.S.); 3Department of Plant Pathology, University of Arkansas, Fayetteville, NC 72701, USA; 4Department of Applied Economics, International Science, Technology Practice and Policy (InSTEPP), 1994 Buford Ave.248E Ruttan Hall Saint Paul, University of Minnesota, Minneapolis, MN 55108-6038, USA; ssenay@umn.edu; 5Department of Sustainable Agriculture, Biodiversity and Ecosystem Management, Nelson Mandela African Institution of Science and Technology, P.O. Box 447, Arusha, Tanzania; tileye.feyisa@nm-aist.ac.tz

**Keywords:** *Maize chlorotic mottle virus*, next-generation sequencing, phylogenetic analysis, MLN prevalence, *Sugarcane mosaic virus*

## Abstract

Sustainable control of plant diseases requires a good understanding of the epidemiological aspects such as the biology of the causal pathogens. In the current study, we used RT-PCR and Next Generation Sequencing (NGS) to contribute to the characterization of maize lethal necrotic (MLN) viruses and to identify other possible viruses that could represent a future threat in maize production in Tanzania. RT-PCR screening for *Maize Chlorotic Mottle Virus* (MCMV) detected the virus in the majority (97%) of the samples (*n* = 223). Analysis of a subset (*n* = 48) of the samples using NGS-Illumina Miseq detected MCMV and *Sugarcane Mosaic Virus* (SCMV) at a co-infection of 62%. The analysis further detected *Maize streak virus* with an 8% incidence in samples where MCMV and SCMV were also detected. In addition, signatures of *Maize dwarf mosaic virus, Sorghum mosaic virus, Maize yellow dwarf virus-RMV* and *Barley yellow dwarf virus* were detected with low coverage. Phylogenetic analysis of the viral coat protein showed that isolates of MCMV and SCMV were similar to those previously reported in East Africa and Hebei, China. Besides characterization, we used farmers’ interviews and direct field observations to give insights into MLN status in different agro-ecological zones (AEZs) in Kilimanjaro, Mayara, and Arusha. Through the survey, we showed that the prevalence of MLN differed across regions (*P* = 0.0012) and villages (*P* < 0.0001) but not across AEZs (*P* > 0.05). The study shows changing MLN dynamics in Tanzania and emphasizes the need for regional scientists to utilize farmers’ awareness in managing the disease.

## 1. Introduction

Maize is the most important cereal crop and a staple food in sub-Saharan Africa. With over 5 million hectares of its land planted with this crop and a per capita consumption of 128 kg, Tanzania is one of the largest producers and consumers of maize in eastern and southern Africa [1]. Maize is intensively cultivated in the southern highland, lake, and northern zones of Tanzania [2]. Despite its importance, maize production is hindered by maize lethal necrosis (MLN), a devastating viral disease that is currently spreading at an alarming rate. The disease was first reported in the Southern Rift Valley region of Kenya in 2011 [3] and Mwanza and Arusha regions in Tanzania in 2012 [4]. MLN has since spread to several other maize-producing African countries, including Rwanda, the Democratic Republic of the Congo, and Ethiopia [5,6,7,8]. The disease is characterized by elongated yellow streaks parallel to leaf veins, chlorotic mottling, and necrosis [9]. The necrotic symptoms occur at different stages of maize development and can lead to 100% crop loss. 

The reported causative agents of MLN in eastern and central Africa are *Maize chlorotic mottle virus* (MCMV) and *Sugarcane mosaic virus* (SCMV) [3,5,6,7,8,9,10]. MCMV is a positive-sense single-stranded RNA virus belonging to the family *Tombusviridae* and has a genome size of 4.4 kb [11]. SCMV is a positive-sense single-stranded RNA virus belonging to the family *Potyviridae* and has a genome size of 9.6 kb [12]. MCMV alone infects maize causing leaf mosaic and moderate stunting symptoms, while co-infection of maize by MCMV and SCMV results in more severe symptoms of stunting, necrosis, and eventually plant death [9]. Other viruses in the *Potyviridae* family such as *Wheat streak mosaic virus* (WSMV) and *Maize dwarf mosaic virus* (MDMV) have also been reported to synergize with MCMV causing MLN [13]. However, these other viruses have not been reported to associate with MLN in eastern and central Africa. Given the complexity of this disease and the economic importance of maize in sub-Saharan Africa, there is a need for a targeted investigation of the possible presence of other MLN-associated viruses apart from MCMV and SCMV in Tanzania.

Sustainable control of plant diseases requires a good understanding of the epidemiological aspects such as the biology of the causal pathogen(s) and the favorable environmental conditions for compatible pathogen-host interactions [14]. The combination of several viral strains makes the disease even more complex because of varying favorable conditions for virulence and transmission of the viruses. The transmission of MLN has been characterized as mainly vector-borne, but also soil-borne and seed-borne [7,15,16]. MCMV, which has mainly been associated with MLN symptoms, is transmitted by thrips [15], rootworms [13,15,17], and beetles [17]. Most cereal *potyviruses* are transmitted by aphids [16]. 

Initial reports of maize lethal necrosis problems in East Africa were based on experiences by farmers and agricultural extension agents, complemented with diagnostics of the disease by international research organizations [3]. An accurate identification of MLN symptoms by the farmers can play a key role in preventing the spread of the disease, as they can mechanically remove the infected plants and/or apply pesticides to control the vectors. Furthermore, farmers can provide useful information about the occurrence and extent of the spread of this and other diseases [18]. To safeguard maize production and food security in sub-Saharan Africa, there is a need to conduct comprehensive research (etiological, epidemiological, and genetic studies) to facilitate the development of effective and sustainable MLN control measures. Recent advances in diagnostic technologies including next generation sequencing (NGS) tools have been useful for the identification of new viruses and their variants in infected plants [10,19]. The objectives of the current study are to contribute to the existing knowledge on the status of MLN in northern Tanzania by characterizing the causative viruses using NGS, to assess farmers’ awareness and experiences on the spread and damage of MLN, to investigate the prevalence of MLN across villages within different agro-ecological zones, and to identify possible other viruses that co-infect maize.

## 2. Results

### 2.1. Screening for MCMV Using RT-PCR

RT-PCR was used to screen for MCMV as a complement to the visual diagnosis of the MLN symptoms in maize. It also provided the basis for selecting representative samples for viral characterization using NGS. Based on the RT-PCR test, the overall percentage of samples with MCMV was 97%. The descending order of the percentages of samples with MCMV across the regions are as follows: Kilimanjaro (100%), Manyara (94%) and Arusha (93%) (Table 1).

### 2.2. Characterization of MLN Viruses Using Next-Generation Sequencing

A total of 48 RNA libraries from maize samples with MLN symptoms (MCMV positive using RT-PCR) collected from Arusha, Kilimanjaro, and Manyara regions of northern Tanzania were constructed and sequenced using the Illumina Miseq platform at the BecA-ILRI Hub in Nairobi, Kenya. A total of 46,361,174 clean reads with an average length of 17–122 bp were produced after removing adaptor sequences and low-quality reads. The reads were assembled and compared against a plant virus database using BLASTN+ and TBLASTX and the resulting data was visualized using Krona [21]. In addition to MCMV and SCMV, several other plant viruses were detected by blast including *Maize streak virus* (MSV)*, Maize dwarf mosaic virus*, *Sorghum mosaic virus* (SrMV)*, Maize yellow dwarf virus-RMV* (MYDV)*,* and *Barley yellow dwarf virus* (BYDV). Further analysis of de novo and reference assemblies showed that no artefacts were introduced during reference assembly, and sequences for MCMV, SCMV and MSV had significant genome coverage while the rest represented short sequence fragments. Co-infection by MCMV and SCMV were detected in 62% of the sequenced samples (Table 1) while MSV had an 8% incidence. The sequencing coverage and depth of the three viruses are shown in Table 2.

Complete genome sequences of MCMV detected in the current study were deposited into NCBI with the following accession numbers: Arusha (MF467384, MF467383, MF467374, MF467389, MF467380, MF467378, MF467381, and MF467375), Manyara (MF467382, MF467379, MF467390, and MF467377) and Kilimanjaro (MF467387, MF467386, MF467388, MF467391, MF467385, MF467392, and MF467376). The genomes were found to be between 4410 to 4432 nucleotides (nt) long with six open reading frames similar to other previously reported MCMV isolates [11,22,23]. Comparison search of the full-length nucleotide sequence against the NCBI database indicated that the virus is very closely related to MCMV isolates from eastern Africa (accession KP851970.3, KP798454.1, KP772217.1, KT250543.1, KT250546.1) sharing 99% nucleotide sequence identity. Phylogenetic analysis of the coat protein (CP) nucleotide sequences of MCMV from this study and existing MCMV isolates showed that the MCMV CP sequences from Tanzania were highly similar to eastern African isolates but different from Nebraska [23] and Kansas [22] isolates (Figure 1).

Assembly of SCMV genome for samples from different regions of Tanzania gave sequences ranging from 9482 to 9575 nt, which were deposited into NCBI with the following accession numbers: Arusha (MF467400, MF467401, MF467404, MF467393, MF467403, MF467402, and MF467399), Manyara (MF467398, and MF467397) and Kilimanjaro (MF467394, MF467395, and MF467396). These genomes are translated via a large polyprotein precursor containing ten mature proteins similar to other viruses of *Potyviridae* family [24]. A comparison of the full-length nucleotide sequences of SCMV isolates from the current study with those that are publicly available at NCBI showed that the virus had a high nucleotide sequence identity (96%–99%) with the very virulent SCMV isolate BD8 from Hebei, China (JN021933.1). Phylogenetic analysis of the CP nucleotide sequences showed that SCMV signatures from the current study were closely related to an isolate from Kenya (JX286701.1) and isolate BD8 from Hebei-China (Figure 2). This shows that SCMV isolates from this study are highly similar, falling under the same genetic cluster (group I). Apart from the unique SCMV isolate from Kenya (JX286701.1), SCMV isolates from other African countries including Rwanda and Ethiopia did not cluster with SCMV isolates from this study.

Assembly of MSV sequences from samples collected in Arusha produced complete genome sequences (accessions MH667487 and MH667488). Comparison search of the full-length nucleotide sequence of MSV against the NCBI database indicated that the virus is very closely related to MSV isolates from Kenya, Mozambique, and Zimbabwe (accessions FJ882094.1, FJ882100.1 and AF329882.1) sharing 99% nucleotide sequence identity.

### 2.3. Prevalence of MLN in Farmers’ Fields in Northern Tanzania

Visual assessments of the farms by trained research assistants showed that the prevalence of MLN in 2015 differed across regions (*P* = 0.0012) and villages (*P* < 0.0001). The highest MLN prevalence was recorded in Kilimanjaro with a mean of 22% symptomatic maize plants, followed by Arusha (14%) and Manyara (10%). The prevalence did not differ across agro-ecological zones (*P* > 0.05; Table 1).

### 2.4. Farmers’ Experiences on MLN Occurrence and the Associated Yield Loss

Farmers from different villages (Table 3) across three regions in Northern Tanzania were interviewed about specific aspects regarding MLN. The majority of the interviewed farmers (98%) cultivated maize every cropping season in each year. Across the three regions, over half of the interviewed farmers (50%–78%) were aware of MLN symptoms. More than half of the farmers (52%) reported to have observed MLN for the first time in their farms in 2013. The occurrence of the disease across the Northern Tanzania regions was reported to be higher (71%) during the long rain seasons than in dry seasons (21%) and short rain seasons (8%). Less than half of the farmers in each region had observed the previously reported MLN vectors such as beetles, rootworms, aphids, and thrips in their farms. The majority of the farmers interviewed (87%) were using certified seeds; however, none of the cultivated varieties was resistant to the disease. Different methods were used by farmers to manage MLN, including rouging, insecticide application, fertilizer application, and weed elimination. However, the disease remained unmanaged.

Based on the farmers who provided information about the occurrence of MLN between 2012 and 2015, MLN was most prevalent in 2014. The high 2014 MLN prevalence was associated with complete maize yield loss for 88% of the interviewed farmers across the three study regions (Table 3). During assessment in farmers’ fields, typical symptoms of MLN were observed. Symptoms included severe mosaic symptoms, leaf margin necrosis (Figure 3a,c), and mild symptoms (Figure 3b). In Mandaka Mnono-Kilimanjaro, maize fields were found to be highly infested by thrips, the major vector of MCMV [25,26]. Aphids and thrips were also observed in other surveyed areas.

## 3. Discussion

The current study began as a response to an alert by farmers about a disease that been spreading fast within the maize growing regions of northern Tanzania. The study confirms that the disease was maize lethal necrosis (MLN). This disease is not new in Tanzania, as its occurrence and the causal organisms have been previously reportedin Arusha and Mwanza [7]. In this paper, we provide information on the occurrence of MLN in a wider geographical area, confirming the presence of the causal viruses for MLN (SCMV and MCMV) and other viruses, which can potentially threaten maize production in Tanzania. The study entailed interviewing farmers’ about their experiences, direct observation of MLN symptoms in the fields and a confirmation of the causal viruses by RT-PCR and next-generation sequencing (Illumina Miseq) analysis of symptomatic maize from Arusha, Manyara and Kilimanjaro regions of Tanzania.

In addition to MCMV and SCMV complete sequences, NGS analysis revealed the occurrence of MSVand signatures of MYDV, MDMV, BYDV, and SrMV that have been previously reported to cause serious infections in maize [27,28,29]. Given that MSV is widely distributed [7,28] and a recent study reported the presence of MYDV in a mixed infection with MLN viruses [30,31], we extrapolate the possibility of evolved interactions. The interactions could besynergistic, helper-dependence, cross-protection, replacement or mutual suppression that can lead to either an increase or decrease in replication and/or transmission of any of the viruses [32,33,34,35]. Mixed infections in many viruses have been reported to cause severe symptoms as well as provide opportunities for recombination events leading to the emergence of new viral species [36,37]. It is not known whether the other viruses detected in this study (except SCMV and MCMV) can affect MLN symptoms. The present study observed a higher incidence of MCMV (97%) with a higher co-infection (62%) (MCMV and SCMV) in symptomatic maize compared to previous reports (62% of MCMV incidence and a co-infection of 51%) [7]. This difference in co-infection could be caused by changes in conditions that favor the pathosystem. There is a need to analyze the cause of these observations and the associated drivers in further laboratory and field tests.

Based on phylogenetic analysis, MCMV isolates identified in this study are highly similar to those from other east African countries (Figure 1). The observed similarity of MCMV isolates from the current study and those from the neighboring countries showsthat combined regional efforts should be applied to control MLN. Currently, initiatives have been taken by the International Maize and Wheat Improvement Center (CIMMYT) and other agricultural research institutes across east Africa to breed for MLN resistant maize varieties. Putative genes and quantitative trait loci (QTL) for MLN resistance have been identified in tropical maize germplasm and will be useful in the breeding efforts [38,39,40]. Interestingly, we observed significant genetic dissimilarities between the SCMV recovered in the current study compared to those from other east African countries retrieved from the NCBI. The observed differences in SCMV could likely indicate that the virus is more sensitive to variations in climate across geographical locations, as pointed out by other authors [7,10,41,42]. Furthermore, SCMV is an RNA virus, and its variations could be caused by inherent replication errors due to the lack of proofreading activity of RNA-dependent RNA polymerases [41,43]. Previous reports show that quarantine measures to control the movement of MCMV in uninfected areas could be more effective than similar measures for controlling SCMV [5]. It is not known whether the observed diversity in SCMV affects MLN and whether this could complicate the management of the disease.

We emphasize the importance for regional scientists to utilize farmers’ degree of awareness in identifying MLN hot spots and facilitate their capacity to collect data from a wider geographical area to enhance better understanding of the disease complex and epidemiology. The findings in the current study revealed that the majority of the farmers across the three regions had observed MLN in four consecutive years since the first report in 2012. The highest MLN prevalence (complete yield loss in 88% of the surveyed farms) was observed in 2014. A similar magnitude of maize yield loss was previously reported in Western Kenya, with the incidence of MCMV increasing between the years 2013 and 2014 [8]. The devastating disease mainly damages small-scale farmers’ field crops, who cannot afford to buy pesticides to control the vectors, and yet their livelihood depends mostly on agriculture. Small-scale farmers contribute to over 80% of Tanzania’s total maize production [2]. Thus, the reported maize yield losses due to MLN indicate a major threat to the per capita income and food security at large in Tanzania, an African country that is struggling to feed its burgeoning population. Given that small-scale farmers cannot afford to buy pesticides and their current methods used to control MLN (e.g., roguing diseased plants) are not very effective, there is a need to look for more sustainable and targeted methods such as breeding for resistant maize varieties. 

Farmers reported a high incidence of MLN in irrigated farms, where continuous maize cropping is practiced throughout the year. As continuous cropping can lead to a build-up and retention of the viral inoculum, there is a need to train farmers on crop rotation methods. Maize could be replaced by other crops (e.g., legumes), as these would not only break the pathogen cycle but also increase dietary diversity. Besides crop rotation, farmers should be encouraged to practice intercropping of maize with crops that are not affected by either MCMV or SCMV. Intercropping creates a vegetation diversity that can divert or repel the vectors of MLN [44]. In the current study, both farmers and the research assistants observed the two main vectors of MLN (thrips and aphids) in farmers’ fields. The presence of vectors enhances a fast spread of most vector-bone plant viruses. Although seed transmission could play a role in the transmission of MLN [7,45,46], farmers should be trained on integrated pest management strategies to sustainably control the vectors and the spread of MLN. Pull and push technology has been developed and validated as an effective tool for controlling insect-pests within east Africa [47]. As the region works towards developing MLN resistant maize, it is paramount to train farmers on how to apply the push and pull, together with other feasible strategies to effectively combat the disease and hence enhance food security in Tanzania.

## 4. Materials and Methods

### 4.1. Study Site and Design

A two-part survey was conducted in Arusha, Kilimanjaro, and Manyara regions in northern Tanzania in 2015. First, because of farmers’ frequent complains of maize crop loss, our research team aimed to establish whether the reported disease was MLN and whether there were other possible viruses that co-infect maize, based on symptoms and molecular diagnostics. Second, the team conducted a survey to assess farmers’ awareness and experiences on MLN including control strategies.

### 4.2. Assessment of Prevalence and Collection of Samples

Based on information on the reported MLN-like symptoms, including leaf mosaic, stunting, yellowing in leaf margins, and necrosis from agricultural extension agents, farms (*n* = 41) were selected for direct visual observation and sampling of infected maize plants. The farms were within villages stratified across the maize producing AEZs in Northern Tanzania (Figure 4). Within each farm, plants with MLN-like symptoms were scouted in quadrants of one-hundred maize plants (three random quadrants of each quarter-acre of the farm). The mean of the counts of the symptomatic plants (%) was considered as the magnitude of the disease in the sampled farm. MLN prevalence across the villages within AEZs and the regions were compared based on the percentages of symptomatic plants in the sampled farms (Table 1). Consequently, maize leaf samples (*n* = 223) of randomly selected plants containing viral-like symptoms were collected across the farms and dried in silica gel prior to laboratory analysis.

### 4.3. Detection of MCMV Using Reverse-Transcription Polymerase Chain Reaction (RT-PCR)

All leaf samples collected were tested for the presence of MCMV using RT-PCR. Total RNA was extracted from the samples (*n* = 223) using a ZR Plant RNA MinPrep^TM^kit (Zymo Research, Irvine, CA, USA) after homogenization using mortar and pestle in liquid nitrogen. The quality of RNA was checked using formamide-denatured agarose gel electrophoresis on 1x TAE [48] and quantified using a Qubit™ 2.0 fluorometer (Thermo Fisher Scientific, Waltham, MA). Complementary DNA (cDNA) synthesis was performed using the Maxima First Strand cDNA synthesis Kit for RT-qPCR (Thermo Fisher Scientific, Waltham, MA) as per manufacturer’ s instructions, followed by PCR. Reactions consisted of 10 µL of 2x One Taq master mix with standard buffer [20 mM Tris-HCl, 1.8 mM NH_4_Cl, 22 mM KCl, 0.2 mM dNTPs, 5% glycerol, 0.06% IGEPAL CA-630, 0.05% Tween 20 and 25 U/mL One Taq DNA Polymerase] (New England Bio Labs, Ipswich, MA), 0.1 µL (10 µM) of the forward (5′-CGCGGCTGACAAGCAAAT-3′) and reverse primer (5′-ACTGGTTGTTCCGGTCTTG-3′) targeting MCMV, 2 µL cDNA and 7.8 µL of sterile water for a final volume of 20 µL. The thermocycler conditions were as follows: an initial denaturation at 95 °C for 3 min followed by 30 cycles of denaturation at 95 °C for 30 sec, annealing at 49.4 °C for 30 sec, elongation at 72 °C for 1 min and a final extension step at 72 °C for 15 min. The PCR products were visualized under UV light on a stained (gel red) 1.2% agarose gel following electrophoresis.

### 4.4. Characterization of MLN Viruses by Next Generation Sequencing

Based on the results of the RT-PCR for MCMV, a subset of samples positive for MCMV from Kilimanjaro (*n* = 15), Arusha (*n* = 22), and Manyara (*n* = 11) were selected for NGS characterization of the MLN causal viruses. The samples (*n* = 48) were used for the construction of sequence libraries using the Illumina TruSeq RNA library prep kit following the manufacturer’s instructions (Illumina, San Diego, CA, USA). Briefly, 0.5µg of RNA/sample was fragmented using the fragmentation mix (Elute, Prime, Fragment High Mix) followed by first strand and second strand cDNA synthesis. The double-stranded cDNA was purified using Agencourt AMPure XP magnetic beads (Beckman Coulter, Indianapolis, IN, USA) followed by end repair and adapters ligation. The ligated ds-cDNA was amplified in a PCR using universal and index primers. Resulting libraries were purified using Agencourt AMPure XP magnetic beads. Libraries’ quality and quantity were assessed with the Agilent Tape Station 2200 system (Agilent Technologies, Santa Clara, CA, USA) and a Qubit™ fluorometer (Thermo Fisher Scientific, Waltham, MA, USA), respectively. The libraries were then normalized to a concentration of 10nM, pooled, and diluted to a final concentration of 6.5 pM. Pooled libraries were sequenced on the Illumina MiSeq system at the BecA-ILRI Hub (Nairobi, Kenya) generating 151 paired-end reads.

### 4.5. Assessment of Farmers’ Awareness and Experiences on MLN

A random group of farmers (*n* = 137) selected with the help of extension staff representing different villages (Lyamungu Kati, Mandaka Mnono, Mlangarini, Ngaramtoni, and Ayasanda) within the three regions of Northern Tanzania were interviewed about specific aspects relating to awareness and experiences of MLN in 2015. Farmers provided information about whether they had observed MLN symptoms (pictures of the symptomatic plants were shown to farmers) in their maize crop within the past four years (2012–2015), the approximate percentage of maize losses due to MLN in comparison with previous yields, seed types and origin used, the predominant insect pests identified by using pictures and maize insect pests identifier from CIMMYT [49], whether they sprayed pesticides to control the insects, and any other methods adopted to manage MLN in their farms.

### 4.6. Data Analysis

The quality assessment of the sequence reads generated was performed using Fastqc v0.11.2 [50]. The FASTX_toolkit [51] and SolexaQA [52] were used for the removal of adapters and poor quality sequences. De novo assembly of the reads was performed using Trinity v2.2.1 [53]. Assembled sequences were then blasted against a locally installed plant virus database using BLASTN 2.2.30+, TBLASTX 2.2.30+ [54]. Identities of the viruses present in each sample were visualized using Krona [21]. Reference mapping was performed for individual samples against the most similar reference genome downloaded from NCBI, using CLC Genomics Workbench 5.5.1 software (Qiagen, Germany).

Multiple sequence alignment of the viruses was performed using CLC Genomics Workbench 5.5.1 software. Nucleotide sequences of MCMV and SCMV coat proteins were used for phylogenetic analysis in Mega 6.0 [55], where a maximum likelihood method based on the Kimura 2-parameter model [56] was used with 1000 bootstrap replicates. Sequences of MCMV, SCMV, and MSV were submitted to the GenBank using the BankIt sequence submission tool [57].

Farmers’ interviews and field observation data were analyzed using JMP Pro v.12 (SAS Institute Inc. 2013). Means of symptomatic plants across villages within the regions and AEZs were computed and compared in a nested linear regression model. 

## 5. Conclusions

This study contributes to a better understanding of MLN in Northern Tanzania, by providing information on disease prevalence across different AEZs in Kilimanjaro, Arusha and Manyara regions in 2015, detection of the associated viruses and indicates possible measures to manage the disease. Furthermore, the study detected other potential viruses that could present a threat to maize production in Northern Tanzania. The complete sequences of the virus isolates reported in this study provide additional resources for the development of diagnostic tools and for enhancing understanding of genetic relatedness of isolates of MCMV, SCMV, and MSV across Africa and their management. Given that the observational and sampling component of the current study were limited in scope, we propose further studies to expand these aspects in order to assess the seasonal variations of the epidemics, the role of the interaction among multiple viruses on severity of MLN, how the genetic variability of SCMV affects MLN, and the role of vectors in respect to changes in the climatic factors.

## Figures and Tables

**Figure 1 pathogens-09-00004-f001:**
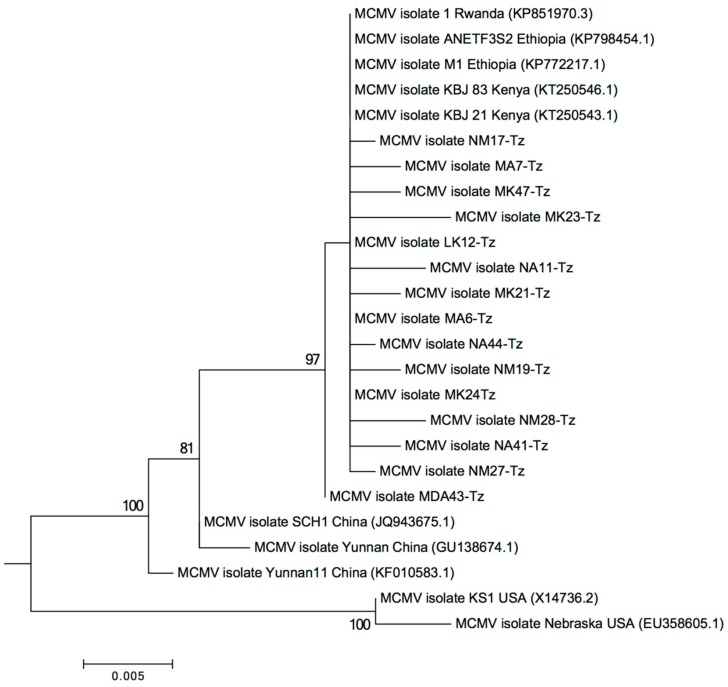
Phylogenetic analysis of the coat protein gene in *Maize Chlorotic Mottle Virus* constructed with MEGA 6.0 using the maximum likelihood method based on the Kimura 2-parameter model with 1000 bootstrap replicates.

**Figure 2 pathogens-09-00004-f002:**
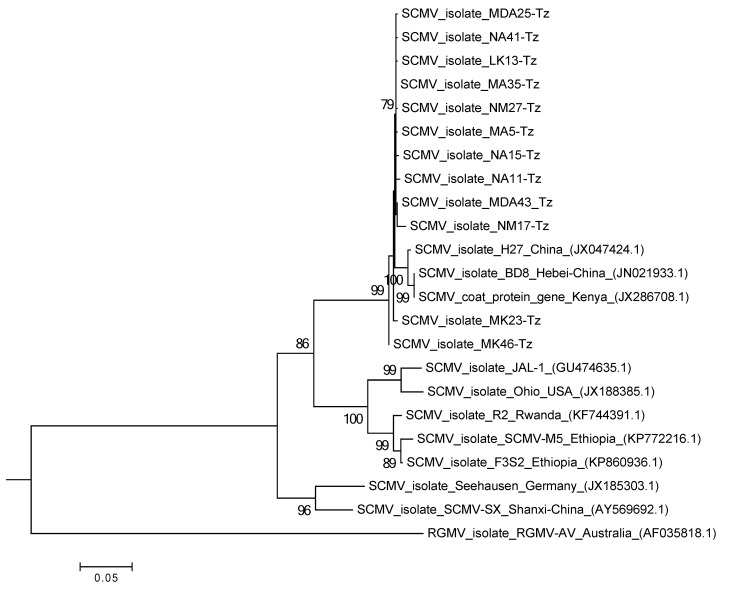
Phylogenetic analysis of the coat protein gene in *Sugarcane mosaic virus* constructed with MEGA 6.0 using the maximum likelihood method based on the Kimura 2-parameter model with 1000 bootstrap replicates. *Ryegrass mosaic virus* (RGMV) was used as an outgroup.

**Figure 3 pathogens-09-00004-f003:**
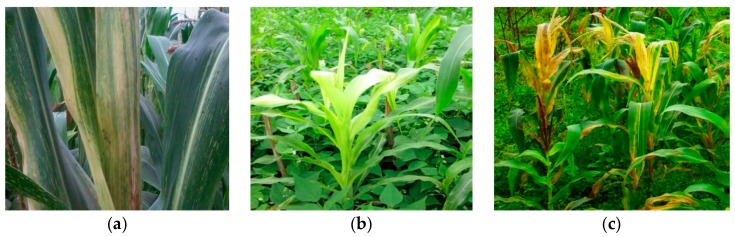
Maize leaves with symptoms of MLN. (**a**) Maize crop from Madira-Arusha with chlorotic mottling, (**b**) maize crop from Mandaka Mnono in Moshi-Kilimanjaro, and (**c**) maize crops from Lyamungu Kati in Hai-Kilimanjaro with dead-heart symptom.

**Figure 4 pathogens-09-00004-f004:**
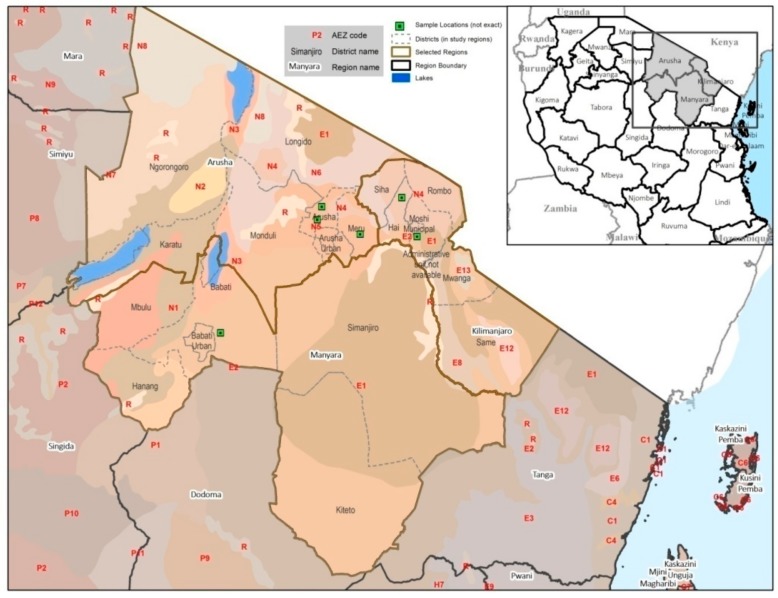
Map showing study areas and their corresponding agro-ecological zones in Northern Tanzania.

**Table 1 pathogens-09-00004-t001:** MLN incidence and prevalence across villages within agro-ecological zones in Northern Tanzania in 2015.

Region	Villages	Agro-Ecological Zones (AEZs) ^a^	Sampled Farms (*n*)	* Maize Plants with MLN Symptoms (%)	Leaf Samples Collected (*n*)	Samples with MCMV (*n*) by RT-PCR	Samples Selected for NGS	Samples with SCMV (*n*) by NGS	Co-infection with SCMV and MCMV (*n*) by NGS
Kilimanjaro	Lyamungu Kati	N4	8	20.6 ± 2.4A	44	44	6	6	6
	Mandaka Mnono	E2	5	24.0 ± 2.9A	31	31	9	0	0
Sub-total 1			13	22.0 ± 1.9A	75	75	15		6
Arusha	Ngaramtoni	N5	14	19.1 ± 1.6A	58	57	8	6	6
	Madira-Sing’isi	N5	3	16.0 ± 3.4AB	35	35	8	8	8
	Tengeru	N5	6	4.7 ± 2.6B	-	-	-	-	-
	Mlangarini	N5	3	2.8 ± 4.2B	20	16	6	6	6
Sub-total 2			26	14.0 ± 1.6B	113	108	22		20
Manyara	Ayasanda	E2	1	10.0 ± 5.2AB	-	-	-	-	-
	Nyunguu	E2	2	9.9 ± 4.2AB	35	33	11	4	4
Sub-total 3			3	10.0 ± 3.3B	35	33	11	4	4
Total			41		223	216 (97%)	48	30	30 (62%)

* Areas connected with common letter A or B do not differ statistically and vice versa. ^a^ E2, N4 and N5 are agro-ecological zones (AEZs) as per the Ministry of Agriculture [20]. E2, N4 and N5 differed in rainfall (800–1000 mm, 500–1400 mm and 600–1200 mm) and altitudes (500–1200 masl, 900–3500 masl and 1300–1700 masl), respectively.

**Table 2 pathogens-09-00004-t002:** Read counts and genome coverage of *Maize Chlorotic Mottle Virus*, *Sugarcane Mosaic Virus* and *Maize Streak Virus* obtained from reference assembly.

Virus	Isolate	Region Collected	Accession Number	Read Mapped	% Read Mapped	Average Depth of Sequence	% Genome Coverage	Genome Length (nt *)
MCMV	MA5-Tz	Arusha	MF467384	578,660	65.7	15,057	99.9	4432
	MA7-Tz	Arusha	MF467383	408,433	82.8	10,668	99.4	4410
	NA11-Tz	Arusha	MF467374	433,159	69.5	11,176	99.7	4422
	LK14-Tz	Kilimanjaro	MF467392	731053	80.4	18,797	99.9	4431
	NA16-Tz	Arusha	MF467375	429,822	37.3	10,795	99.9	4431
	NM19-Tz	Manyara	MF467382	466,171	38.9	11,955	99.8	4428
	MK21-Tz	Kilimanjaro	MF467385	480,071	50.1	11,738	99.5	4416
	MK23-Tz	Kilimanjaro	MF467376	710,295	52.9	16,979	99.9	4431
	NM27-Tz	Manyara	MF467379	548,930	64.9	13,866	99.8	4427
	NM28-Tz	Manyara	MF467390	442,863	40.2	11,291	99.8	4429
	LK12-Tz	Kilimanjaro	MF467387	453,118	73.7	11,767	99.75	4425
	MK34-Tz	Kilimanjaro	MF467386	498,579	49.2	12,741	99.8	4428
	MK24-Tz	Kilimanjaro	MF467388	663,071	73.4	16,529	99.9	4431
	MK47-Tz	Kilimanjaro	MF467391	38,431	2.04	928	99.7	4423
	NM17-Tz	Manyara	MF467377	496,273	59.3	12,672	99.5	4415
	MA6-Tz	Arusha	MF467389	397,723	75.7	10,425	99.5	4416
	NA41-Tz	Arusha	MF467380	448,926	58.5	11,521	99.9	4432
	NA44-Tz	Arusha	MF467378	657,532	60.3	16,422	99.6	4418
	MDA43-Tz	Arusha	MF467381	452,173	38.5	11,294	99.9	4431
SCMV	MK23-Tz	Kilimanjaro	MF467394	27,976	2.1	309	99.4	9522
	MDA43-Tz	Arusha	MF467400	14,677	1.3	168	100	9575
	MK46-Tz	Kilimanjaro	MF467395	11,321	0.8	131	99.3	9511
	NA15-Tz	Arusha	MF467402	17,531	2.0	209	99.0	9484
	NA41-Tz	Arusha	MF467399	18,657	2.4	211	99.1	9491
	NM27-Tz	Manyara	MF467398	13,241	1.6	152	99.0	9482
	NA11-Tz	Arusha	MF467393	9473	1.5	113	99.4	9520
	MA35-Tz	Arusha	MF467403	9292	1.8	115	99.5	9527
	NM17-Tz	Manyara	MF467397	9743	1.5	115	99.4	9520
	MDA25-Tz	Arusha	MF467401	14,343	1.9	164	99.1	9492
	MA5-Tz	Arusha	MF467404	11,621	1.3	137	99.1	9494
	LK13-Tz	Kilimanjaro	MF467396	10,510	1.4	125	99.1	9487
MSV	NA15-Tz	Arusha	MH667487	8937	0.4	377	100	2689
	MDA26-Tz	Arusha	MH667488	992	0.1	37	100	2689

* nt = nucleotide.

**Table 3 pathogens-09-00004-t003:** Farmers’ awareness and experiences about MLN across villages within agro-ecological zones in Northern Tanzania in 2015.

Region	Villages	Agro-Ecological Zones (AEZs)	Interviewed Farmers (*n*)	Farmers had Recognized MLN in Their Farms (%)	Farmers Observed Known Insect-Vectors of MLN (%)	Farmers Reported Complete Maize Yield Loss Due to MLN in 2014 (*n*)
Kilimanjaro	Lyamungu Kati	N4	29	59	48	29
	Mandaka Mnono	E2	24	67	17	23
**Sub-total 1**			53			52
Arusha	Ngaramtoni	N5	27	78	30	25
	Mlangarini	N5	30	50	27	19
**Sub-total 2**			57			44
Manyara	Ayasanda	E2	27	67	22	25
**Sub-total 3**			27			25
**Total**	-		137			121

N4, E2, and N5 are agro-ecological zones as per the Ministry of agriculture [20].

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
