# Peer review of "Status and Epidemiology of Maize Lethal Necrotic Disease in Northern Tanzania"

_pathogens, 2019, doi:10.3390/pathogens9010004_

Round 1

Reviewer 1 Report

I have no further comments to the paper.

Author Response

We would like to acknowledge the inputs provided towards improving our manuscript.

Reviewer 1 had no further comments on the current version.

Reviewer 2 Report

The authors raise a very important problem of the destruction of maize crops in Tanzania by viruses causing maize lethal necrotic. Due to the fact that maize is the basic food in this country, this problem has attracted particular attention of farmers and scientists in recent years. Much attention was paid to diagnosing the problem, and unfortunately a smaller part of the manuscript indicates the possibilities of solving it. Detailed comments are presented below.

p. 1, l. 2 epidemiology > Epidemiology

p. 1, l. 2 lethal > Lethal

p. 1, Abstract (adjust the Abstract to left and right margins)

p. 1, l. 24 generation > Generation

p. 1, l. 25 sequencing > Sequencing

p. 2, l. 6 mottling, and > mottling and

p. 2, l. 36 epidemiological, and > epidemiological, and

p. 2, l. 50 Kilimanjaro, (100%) Manyara > Kilimanjaro (100%), Manyara

p. 3, Table 1 (Table 1 cannot go out on the margins)

p. 3, l. 5 Rainfall > rainfall

p. 3, l. 6 masl) respectively. > masl), respectively.

p. 3, l. 13 TBLASTX, and > TBLASTX and

p. 3, l. 16 (MYDV), and > (MYDV) and

p. 4, Table 2 (Table 2 cannot go out on the margins)

p. 5, l. 18 nt which > nt, which

p. 5, l. 20 MF467397)and > MF467397) and

p. 5, l. 20 MF467395, and > MF467395 and

p. 6, l. 6 that, SCMV > that, SCMV

p. 6, l. 9 study. . > study.

p. 6, l. 9 Figure 2.Phylogenetic > Figure 2. Phylogenetic

p. 6, l. 23 (10%) (Table 1). > (10%). (this table is referenced after next sentence)

p. 7, l. 6 farm > farms

p. 7, l. 17 agriculture[20]. > agriculture [20].

p. 7, l. 19 in the year 2014 > in 2014

p. 8, l. 9 information is provided for > provided the information on

p. 8, l. 9/10 a confirmation of > confirming the presence of

p. 8, l. 10 viruses which > viruses, which

p. 8, l. 13 generation sequencing > Generation Sequencing

p. 8, l. 14 Tanzania.. > Tanzania.

p. 8, l. 14 In addition (this sentence, please start with a new paragraph)

p. 8, l. 26 62% MCMV > 62% of MCMV

p. 8, l. 26 51%)[7] > 51%) [7]

p. 8, l. 33 (CIMMYT)and > (CIMMYT) and

p. 8, l. 41 polymerases[41,43] > polymerases [41,43]

p. 9, l. 26 aphids)in > aphids) in

p. 9, l. 31 MLN. . > MLN.

p. 9, l. 39 in the year 2015 > in 2015

p. 9, l. 46 farms(n > farms (n

p. 10, l. 8 4.Map > 4. Map

p. 11, l. 9 MA) respectively > MA), respectively

p. 11, l. 34 [55] where, a > [55], where a

p. 11, l. 43 in > in (remove underline)

p. 12, l. 24 study; in > study, in

p. 12, l. 24 data; in > data, in

p. 12, l. 24 manuscript, or > manuscript or

p. 12, l. 25 results”. > results.

References (please shorten the doi number, f. ex. for the first reference: https://doi.org/10.13031/aim.20152189434 > doi: 10.13031/aim.20152189434

Author Response

Dear Reviewer,

We would like to acknowledge the comments provided towards improving our manuscript.

This manuscript is a resubmission of an earlier submission. The following is a list of the peer review reports and author responses from that submission.

Round 1

Reviewer 1 Report

General comments

The article entitled “Status and epidemiology of Maize lethal Necrotic Disease in Northern Tanzania” focus on the identification of the viruses and on the spread of MLN in northern Tanzania. MLN is a result of a combination of two viruses, the Maize Chlorotic Mottle Virus (MCMoV) and any of the cereal viruses in the Potyviridae group, like the Sugarcane Mosaic Virus (SCMV). The authors have used RT-PCR and NGS techniques to identify the viruses on a large number of maize samples from different areas of Tanzania. The authors have also interviewed farmers to get their perception on how the disease is spreading and the effects that it has on maize production. The disease can be devastating and farmers can lose their whole production when affected. This dual approach makes sense as farmers can provide useful information about the occurrence and extent of spread.

Maize makes up a large part of the diet in East Africa, and is a staple crop food around the world. Therefore, this study is relevant by providing useful genomic data towards a better management of the disease.

I think that the paper is well structured, and easy to follow. I think however that the language could be greatly improved.

Minor comments:

Lines 23-25: Please change sentence to: “In the current study, we have used RT-PCR and Next generation sequencing (NGS) to contribute to the characterization of MLN viruses and to identify other possible viruses that could represent a future threat in maize production in Tanzania.”

Line 29: “…symptoms can play a key role…”

Line 31: “Furthermore, farmers can provide…”

Line 44: The sentence is confusing. Please rephrase.

Page 5, line 1: Please change sentence to: “The complete genome sequences of MCMV were retrieved from NCBI using the accessions:……”

Page 6, line 2: Please change sentence to: “…showed that the virus had a high nucleotide sequence identity (96-99%) with the highly virulent SCMV….”

Page 6, line 9: remove “showing variations”

Page 6, line 20: remove comma

Page 8, line 10: please remove “of the generated reads”

Page 8, lines 12-13: the sentence is confusing. Please change to “….and a recent study reported the presence of….”

Page 8, lines 21-22: the sentence is confusing, please rephrase.

Page 8, line 26: remove comma

Page 8, line 31-32: Please rephrase

Page 8, line 35: please remove “in turn, variations” or rephrase. I understand that the authors want to say that the errors can lead to variations, but the sentence is unclear.

Page 9, line 2: change to: “revealed that the majority of the farmers across regions had observed MLN in four consecutive years since the first….”

Page 9, line 4: “incidence was found to…”

Page 9, lines 9-14: you use too much “hence”. Please rephrase.

Line 9: “…the farms and dried in silica gel prior to laboratory analysis”

Page 11, line 36: “This study contributed to increase the knowledge on MLN in Northern Tanzania, including the prevalence…”
